# Sources, Pollution Characteristics, and Ecological Risk Assessment of Steroids in Beihai Bay, Guangxi

**Chaoxing Ren [1], Xiao Tan [2], Cuimei Huang [1], Hui Zhao [2,\*] and Wenlu Lan [1,\*]**

[1]  Marine Environmental Monitoring Center of Guangxi, Beihai 536000, China; rencx1993@163.com (C.R.); summer3807892511@163.com (C.H.)

[2]  School of Chemistry and Environment, Guangdong Ocean University, Zhanjiang 524088, China; tanxiao3@mail3.sysu.edu.cn

\*  Correspondence: huizhao1978@163.com (H.Z.); dr.lan@139.com (W.L.); Tel.: +86-13729027361 (H.Z.)

**Abstract:** Steroids are environmental endocrine disruptors that are discharged from vertebrates and are also byproducts of aquaculture. They have strong endocrine disrupting effects and are extremely harmful to the environment. The pollution of steroids in Beihai Bay was assessed through analyzing sources from rivers entering the bay. Six different types of steroids were detected in seagoing rivers, seagoing discharge outlets, and marine aquaculture farms, ranging from 0.12 (methyltestosterone) to 2.88 ng/L (estrone), from 0.11 (cortisol) to 5.41 ng/L (6a-methylprednisone (Dragon)), and from 0.13 (estradiol) to 2.51 ng/L (nandrolone), respectively. Moreover, 5 steroids were detected in 13 of the 19 seawater monitoring stations, accounting for 68.4% of the samples, and their concentrations ranged from 0.18 (methyltestosterone) to 4.04 ng/L (estrone). Furthermore, 7 steroids were detected in 15 of the 19 sediment monitoring stations, accounting for 78.9% of the samples, with concentrations ranging from 26 (estrone) to 776 ng/kg(androsterone). Thus, the main source of marine steroids were the discharging rivers and pollution sources entering the sea. An ecological risk assessment indicated that estrone and methyltestosterone were at high risk in this region; 17β estradiol (E2β) was medium risk, and other steroids were of low or no risk. This study provides a scientific basis for ecological risk assessment and control.

**Keywords:** steroid; estrogen; androgens; progesterone; glucocorticoid; Beihai Bay; pollution characteristics; ecological risk assessment

## 1. Introduction

Steroids are bioactive tetracyclic aliphatic hydrocarbon compounds synthesized from cholesterol. According to their biological function, they can be divided into sex hormones including estrogens, androgens and progesterone, and adrenal cortical hormones including glucocorticoids and mineralocorticoids [1]. Steroids in the environment are mainly derived from human and animal excreta (urine and feces) and from the use and release of synthetic steroids. As endocrine disruptors, steroids can cause serious harm to the reproductive, nervous, and immune systems of organisms at very low concentrations (ng/L level), leading to the imbalance of endocrine systems; in this context, they attract extensive attention from the public and the scientific community [2–5]. Studies have revealed high concentrations of steroids in sewage treatment plants [6–8], livestock farms [9,10], aquaculture farms [11], sludge [12], rivers [13–15], and soil [16] in China. There are also some studies addressing the pollution characteristics of steroids in marine environments and organisms [2,12,17–28].

The Beibu Gulf is an important hub connecting China and Southeast Asian countries in commerce and trade, and also an important node for China to develop the Belt and Road Initiative and Maritime Silk Road. In recent years, with the deepening of the China–ASEAN strategic cooperation and ongoing development in western China, industry and aquaculture in the Beibu Gulf region have developed rapidly. However, a large

number of pollutants discharged into the environment have adversely affected the offshore water quality. Beihai is the core city of the Beibu Gulf. Its economy has grown rapidly as a result of its port and marine fishing industry. Beihai Bay, which includes Lianzhou Bay and Tieshan Port Bay, and belongs to the Beibu Gulf, Guangxi, is located in South China's Guangxi Zhuang Autonomous Region. It is a well-known tourist resort and has a subtropical monsoon climate. As a typical semi-closed environment, Lianzhou Port Bay and Tieshan Port Bay receive various pollutants from terrestrial wastewater via municipal sewage, aquaculture wastewater, and discharging rivers. In recent studies, heavy metals [21,22], persistent organic pollutants such as polycyclic aromatic hydrocarbons [23], polychlorinated biphenyls [24], organochlorine pesticides [25], and new pollutants such as phenols [26] and antibiotics [27] have been detected in sediments or in the water environment of the Beibu Gulf. However, the sources and pollution characteristics of the steroids in the marine environment of Beihai Bay have not been investigated, and a risk assessment has not yet been conducted. This study can act as a reference for the government to establish a standardized steroids monitoring and management system.

## 2. Materials and Methods

### 2.1. Chemicals and Sample Collection

A number of 28 high-purity natural and anabolic steroids were purchased from multiple chemical suppliers, including: 9 androgens (trenbolone (TBL), androsterone (ADS), nondrolong (19-NT), boldenone (BOL), epitestosterone (ET), testosterone (TTR), methyl estosterone (MT), trenbolone-acetate (TLA), and stanozolol (SZL)); 5 estrogens (estrone (E1), 17-beat-estradiol (E2$\beta$), 17-alpha-estradiol (E2$\alpha$), estriol (E3), and 17A-ethinylestradiol (EE2)); 7 glucocorticoids (cortisol (CRL), meprednisone, prednisolone (PREL), 6-alpha-methylprednisolone (MPREL), betamethasone (BET), beclometasone, fluticasone propionate); and 7 progestogens (gestrinone (GR), gestodene, norgestrel (NGT), progesterone (P), medroxyprogesterone (MP), megestrol-17-acetate (MGA), and medroxyprogesterone-acetate (MPA)). More details about the steroids are shown in Table S1.

The study area included five major river systems entering the sea, namely, the Nanyu (R1), Yaqiao (R2), Ximen (R3), Nankang (R4), and Baisha (R5). In 2020, a number of illegal sewage outlets and integrated sewage outlets were shut down or intercepted during a special campaign to comprehensively control sewage outlets in the Guangxi coastal area. Five municipal sewage treatment plants and a wharf-integrated sewage outlet were the main marine pollution sources in the present study area, including Hongkan sewage treatment plant (S1), Dangjiang sewage treatment plant (S2), Xichang town sewage treatment plant (S3), TieShan Port sewage treatment plant (S5), Yingpan town sewage treatment plant (S6), and Dijiao wharf (S4). Five representative and scaled aquafarms (numbered A1–A5) from Tieshan Port and Hepu were selected for the sampling of the aquaculture wastewater, as there are many aquafarms distributed in these areas. In addition, a total of 19 monitoring sites (numbered W1-W19) were set up in Beihai Bay and the surrounding coastal areas.

From 2 July to 25 September 2021, a total of 19 seawater samples, 19 sediment samples, 5 river water samples, 10 domestic sewage samples including the import and export of domestic sewage treatment plants, and 5 aquaculture wastewater samples were collected (see Figure 1 for sampling locations). At least 10% were determined as replicates or blanks for each batch of samples. Each 2L seawater sample was collected from 0.1 to 1.0 m below the surface at low tide on the sampling date. On the sampling date, the dilution effect was minimal, and the actual pollution of the marine environment caused by steroids could be better reflected. The estuary samples were all collected at low tide, when the sea water receded and the salinity of the river was less than 2%, to represent the surface water. The sample containers were cleaned with methanol and Milli-Q water, and used to collect a test samples successively; then, 100 mL of methanol was immediately added to each water sample and the pH of the sample was adjusted to 3 with 4 M $H_2SO_4$. Surface sediment (1–2 kg) from the top 0–30 cm was collected by a stainless steel grab and placed in a sealed aluminum box. All collected samples were immediately placed in an incubator and kept at

4 °C. They were then transported to the laboratory and the pre-treatment was completed within 48 h. After freeze-drying and grinding, the solid samples were screened through a 60-mesh filter and stored at 4 °C for subsequent analyses.

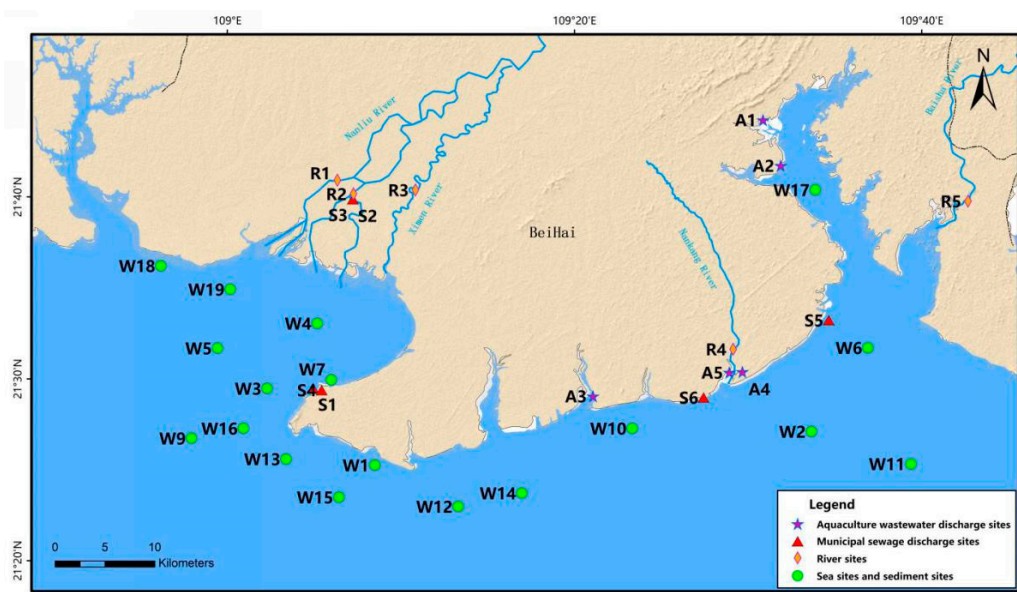

**Figure 1.** Map of water and sediment sampling stations in Beihai Bay, Guangxi.

### 2.2. Sample Extraction and Instrumental Analysis

Sample extraction and instrumental analysis were performed according to the studies of Liu et al. [2,12]. In brief, a concentration of 0.1 mg/L internal standard 100 μL was added to each water sample (1–2 L), and the extraction of analytes was performed by a solid phase extraction (SPE) technique, using a Waters Oasis HLB cartridge (500 mg, 6 mL). Methanol (10 mL) and ethyl acetate (10 mL) were successively used for elution, then nitrogen was used to blow-dry, and dilution took place with methanol to a volume of 1 mL, followed by filtering through a 0.22 μm organic phase needle membrane, before being set aside until needed. The 5.0 g sediment samples were extracted with ethyl acetate/methanol (1/1, *v/v*) using an accelerated solvent. The extraction conditions were as follows: heating equilibrium 3 min; extraction temperature 100 °C; extraction pressure 10.3 MPa; extraction 5 min for two cycles. The extraction liquid was concentrated by rotary evaporation at 40 °C until nearly dry, and then 10 mL water was added and shaken to obtain the extracted solution. The above solution was purified by HLB column, and eluted with 10 mL of methanol and 10 mL ethyl acetate, successively. Subsequently, nitrogen was used to blow-dry the samples, and diluting took place with methanol to a volume of 1 mL, followed by filtering through a 0.22 μm organic phase needle before being set aside until needed.

An ACQUITY UPLC ultra performance liquid chromatograph and a Xevo TQ-S triple quadrupole tandem mass spectrometer (WATERS, Corp. Milford, MA, USA) were used to determine the target compound. The multiple response monitoring (MRM) mode was used for the quantitative analysis of target compounds, with isotope-labeled compounds as internal standards. The chromatographic column used Waters ACQUITY UPLC BEH C18 (50 mm × 2.1 mm, 1.7 μm); the column temperature was 40 °C, and the injection volume was 1 μL. Liquid mobile phase gradient, ratio, and flow rate are shown in Table S2, and mass spectrometry conditions are shown in Table S1.

### 2.3. Quality Control

For the standard recovery experiment using an internal standard method, 28 steroids and 4 internal standards, the same as the target compound and isotope markers, were added to ultrapure water (1 L) and clean, freeze-dried sediment samples (5 g). The recovery rate, method detection limit, and lower limit of quantification were calculated to measure

the reliability of the method. Three experiments with different concentrations (10, 50 and 100) were conducted in ultrapure water (ng/L) and sediment (ng/g). Low-concentration extraction with internal standards was performed on the ultrapure water and sediment, the signal-to-noise ratio was calculated, and the concentration was divided by the signal-to-noise ratio. The detection limit of this method was 3, the lower limit of quantification was 10.

### 2.4. Method Validation

The detection limit, lower limit of quantification, and recovery rate of the 28 steroids from the samples are shown in Table S3. The detection limit and the lower limit of quantification for the ultrapure water method were 0.01–3.43 ng/L and 0.04–11.42 ng/L, respectively. For the sediment method, these values were 0.02–5.59 ng/g and 0.13~18.64 ng/g, respectively. The recoveries of steroids were all within the acceptable range, among which the recovery rate of ultrapure water was 83.7–118%, and the recovery rate of sediment was 81.5–128%. In general, the method successfully implemented the extraction and detection of 28 steroids from the water and sediment.

## 3. Results and Discussion

### 3.1. Distribution of Steroid Levels

#### 3.1.1. Distribution of Steroid Levels in Estuaries

Steroids were detected in all of the monitored sections of the five rivers. Six steroids were detected, including three estrogens (estrone, 17β -estradiol, estriol), two androgens (methyltestosterone, stanozolol), and one progesterone (gestrinone), with concentrations ranging from 0.12 (methyltestosterone) to 2.88 ng/L (estrone). The detection rates of estrogens, androgens, and progesterone were 100%, 80% and 40%, and the maximum concentration were 2.88 ng/L (estrone), 2.60 ng/L (methyltestosterone), and 0.64 ng/L (gestrinone), respectively. No glucocorticoids were detected in the above-monitored sections (shown in Figure 2). This finding is consistent with the levels of steroid concentrations in surface water reported in the literature [28,29].

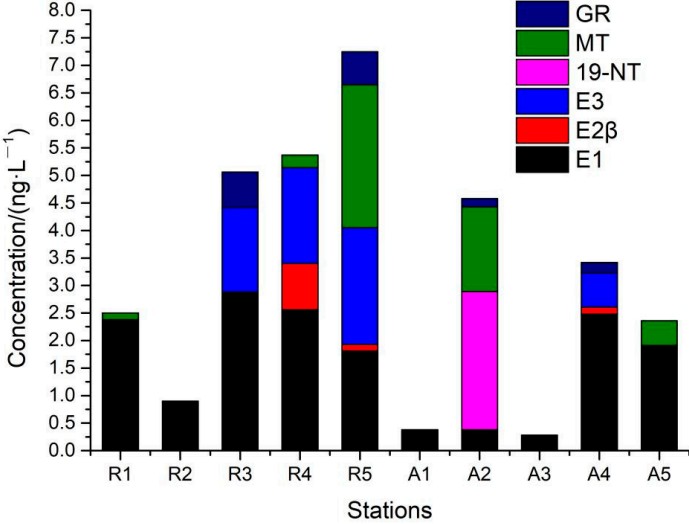

**Figure 2.** Distribution of steroids in seagoing rivers and aquaculture wastewater.

#### 3.1.2. Distribution of Steroid Levels in Sewage Outlets into the Sea

Steroids were detected in the inlets of all five municipal sewage treatment plants, and the maximum concentration was 423.93 ng/L (total). Twelve steroids, including four estrogens, five androgens, one progesterone, and two glucocorticoids were detected with concentrations ranging from 0.32 ng/L (cortisol) to 162.51 ng/L (androsterone). The detection rates of estrogens and androgens were 100% and 80%, and the maximum concen-

trations were 77.84 ng/L (estrone) and 162.51 ng/L (androsterone). Only one progesterone was detected at a concentration of 1.11 ng/L (gestrinone). One glucocorticoid was also detected at a concentration of 7.28 ng/L (6A-methylprednisolone). In general, the inlet water samples contained several to several hundred types of progesterone and glucocorticoid residues at the concentration level of ng/L (shown in Figure 3). This is comparable with the levels of steroids detected in the inlets of wastewater treatment plants in Guangdong and Beijing, China, Japan, and Canada [7,12,13,30].

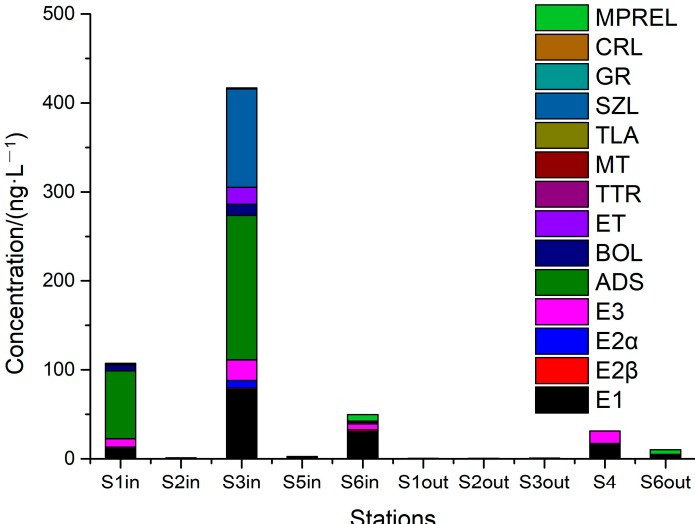

**Figure 3.** Distribution of steroids in the inlets and outlets of municipal sewage treatment plants.

A complete range of steroids were also detected in the outlets of five municipal sewage treatment plants, as well as Dijiao wharf, with a maximum detected concentration of 10.35 ng/L (total). There were two estrogens (estrone, estriol), two androgens (methyltestosterone, stanozolol), and two glucocorticoids (cortisol and 6A-methylprednisolone) detected, with concentrations ranging from 0.11 ng/L (cortisol) to 5.41 ng/L (6A-methylprednisolone). The detection rate of estrogens, androgens, and glucocorticoids were 100%, 40.0% and 40.0%, and the maximum concentrations were 17.19 (estrone), 0.25 ng/L (stanozolol), and 5.41 ng/L (6A-methylprednisolone), respectively. Progesterone was not detected.

Cortisol and 6A-methylprednisolone had the lowest efficiency of removal, at only 25.7%, with estrone ranging from 47.7% to 99.2%; however, 17β-estradiol and 17-α estradiol were completely removed, which indicates that the sewage treatment plants have a certain treatment efficiency for steroids. After treatment, most of the steroids were removed and the concentrations in the outlet water were relatively low, ranging from none detectable (ND) to several ng/L. However, the levels of estriol in the outlet water (1.62 ng/L) were two-fold higher than that of the inlet water (0.8 ng/L), and cortisol (0.77 ng/L) was also 2.4-fold higher; this might be owing to the dissociation of conjugated steroids into a free state, or the conversion from other steroids [31].

### 3.1.3. Distribution of Steroids in Aquaculture Wastewater

Steroids were detected in the wastewater from five aquafarms, with a detection rate of 100%. There were six steroids detected, including three estrogens (estrone, 17β-estradiol, estriol), two androgens (nandrolone, methyltestosterone), and one progesterone (gestrinone), with concentrations ranging from 0.13 ng/L (estradiol) to 2.51 ng/L (nandrolone). The detection rate of estrogens was 100%, and the maximum concentration was 2.48 ng/L (estrone). The detection rate of androgens and progesterone were the same at 20.0%, while the maximum concentrations were 2.51 ng/L (nandrolone) and 0.19 ng/L (gestrinone), respectively. Both nandrolone and getrienone are synthetic steroids, indicating that these were intentionally added to either feed or the aquafarm environment.

### 3.1.4. Distribution of Steroids in Seawater and Sediment

Steroids were detected in 13 of the 19 seawater monitoring stations, accounting for 68.4% of the samples. Five steroids were detected, including one estrogen (estrone), two androgens (methyltestosterone, stanozolol), one progesterone (gestrinone), and one glucocorticoid (cortisol), with concentrations ranging from 0.18 ng/L (methyltestosterone) to 4.04 ng/L (estrone). The detection rate of estrogens, androgens, and progesterone were 68.4%, 47.3% and 21.0%, and the maximum concentrations were 4.04 ng/L (estrone), 1.95 ng/L (methyltestosterone), and 0.8 ng/L (gestrinone), respectively. Glucocorticoids were detected at only one station, with a concentration of 0.26 ng/L (cortisol). These results are shown in Figure 4. These findings are in the same order of magnitude as the concentrations of estrogen detected by Wu Shimin et al. [32] in Liaodong Bay, and the concentrations of androgen, progesterone, and glucocorticoid detected by Liu et al. [2] in Hailing Bay and by Yang Lei et al. [20] in Liusha Bay (South China Sea).

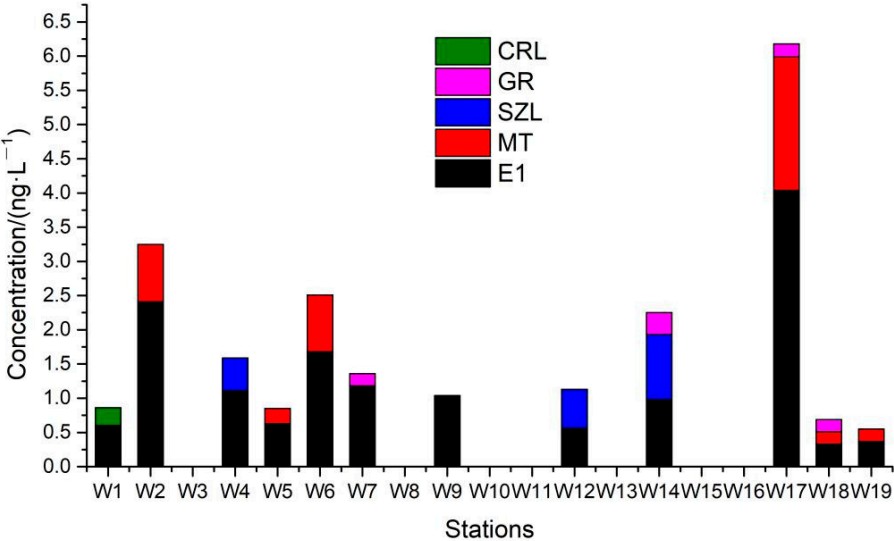

**Figure 4.** Distribution of steroids in seawater sampling stations.

Steroids were detected in 15 of the 19 sediment monitoring stations, accounting for 78.9% of the samples. Among the seven steroids detected in the sediment were three estrogens (estrone, 17β-estradiol, estriol), one androgen (androsterone), one progesterone (progestone), and two glucocorticoids (cortisol, prednisolone), in concentrations ranging from 26 μg/kg (estrone) to 776 μg/kg (androgen). The detection rate of estrogens, androgens, progesterone, and glucocorticoids were 82.4%, 11.8%, 88.2%, and 41.2%, and the maximum concentrations were 406 μg/kg (estrone), 776 μg/kg (androsterone), 776 μg/kg (progestone), and 176 μg/kg (prednisolone), respectively (shown in Figure 5). The estrogen level was similar to previous results found in the Pearl River Delta [33] and Xiamen Bay [34].

### 3.2. Spatial Distribution and Source Analysis of Steroids

The steroids were mostly concentrated in Lianzhou Bay and Tieshan Port Bay of the Beihai Bay area; they were particularly concentrated in Lianzhou Bay, owing to the afflux of the Nanyu, Yaqiao, and Ximenjiang rivers and sewage from the Hongkan, Dangjiang, and Xichang sewage treatment plants of Beihai city and Dijiao wharf. Meanwhile, the weakness of dilution ability and the low water exchange rate in Lianzhou Bay mean that steroids are difficult to degrade in this area [35,36], probably leading to the higher level of steroids in the sediment when a large number of steroids move from terrestrial sources into the sea.

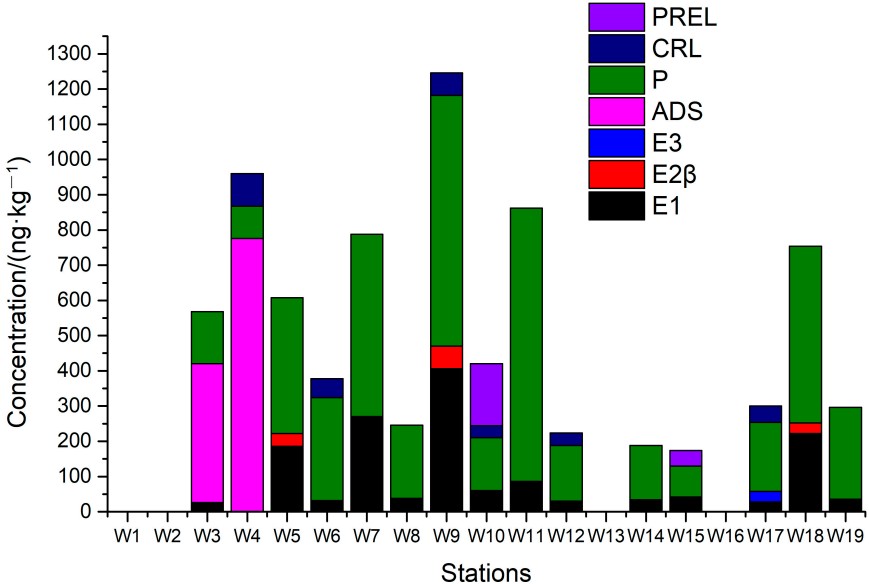

**Figure 5.** Distribution of steroids in sediment sampling stations.

The steroids with detection rates of more than 10% were estrone (estrogen), progesterone (progestone), methyltestosterone (androgen), estriol (estrogen), 17β estradiol (estrogen), cortisol (glucocorticoid), stanozolol (androgen), and gestrinone (progesterone), in descending order. Gestrinone, a synthetic steroid, was detected in all the different types of monitoring stations, indicating that human activities have increased the movement of pollution from rivers and sewage plants to the coastal area. It is worth noting that progesterone can affect and damage the physiological status and reproductive development of fish [3,37,38].

Nandrolone, another synthetic steroid that is used as an incitant to treat refractory anemia, and a variety of wasting diseases generating the loss of weight and muscle, was detected in aquaculture wastewater. It has strong poisonous side effects and can cause cancer, hepatitis, and nervous disorders. Prednisolone and other glucocorticoids are widely used in veterinary care to suppress various allergic, inflammatory, and autoimmune diseases. Comparatively, the detection rate of glucocorticoids in the water environment was lower than for other steroids. For example, prednisolone was detected only at the outlet of municipal sewage plants, suggesting it as the primary source for the deposits found in marine sediment. A large number of steroids at high concentrations were detected from the inlets of sewage treatment plants, revealing that untreated municipal sewage is a major source of pollution to the marine environment. Untreated sewage from coastal cities has been directly discharged into the sea for many years, which has resulted in an increase in steroid levels in the sediment of Beihai Bay. As the above results show, the steroids detected in the water samples were influenced not only by intentional addition, but also by the behavior of the surrounding environments.

*3.3. Ecological Risk Assessment*

3.3.1. Ecological Risk Assessment of Steroids in a Water Environment

According to the European Commission Technical Guidance Document, a preliminary screening-level risk assessment can be established using the RQ approach [39].

$$RQ = \frac{MEC}{PNEC} \tag{1}$$

where *MEC* (measured environmental concentration) and *PNEC* (predicted no-effect concentration) are used to calculate the RQ value. The *PNEC* value is calculated as follows:

$$\text{PENC} = \frac{\text{L(E)C}_{50} \text{ or NOEC}}{\text{AF}} \tag{2}$$

where $LE_{50}$ or $LC_{50}$ are acute toxicity factors, while NOEC is the chronic toxicity factor, and AF comprises the assessment factors. When $\text{L(E)C}_{50}$ data were used, AF was 1000, and when NOEC data were used, the AF value was 100. The RQ value is used to represent the degree of ecological environmental risk. The higher an RQ value, the higher the risk of pollutants in the water environment. Among them, an RQ value ranging from 0.01 to 0.1 indicates low risk, 0.1 to 1 indicates moderate risk, and >1 indicates high risk [2]. If sufficient toxicity data are available, the *PNEC* value can be derived using the method of species sensitivity distribution [40].

Currently, there are plenty of data regarding the aquatic toxicity of estrogen, but few *PNEC* values have been reported for other steroids, especially androgens, progesterone, and glucocorticoids. According to the relevant literature [20,41,42], seven *PNEC* values of steroids with limited toxicity data, including estrone, 17β-estradiol, estriol, methyltestosterone, stanozolol, gestrinone, and cortisol were calculated using assessment factors in this study. The *PNEC* values of stanozolol and gestrinone were defined by the median of the chronic toxicity (NOEC) of daphnia and fish under high and medium concentrations (0.1 mg/L); the RQ values shown in Table 1 were calculated from them.

**Table 1.** RQ value of target steroids in the water environment.

| Station | Estrone | 17β-Estradiol | Estriol | Methyltestosterone | Stanozolol | Gestrinone | Cortisol |
|---------|---------|---------------|---------|---------------------|------------|------------|----------|
| W1 | 0.2 | 0 | 0 | 0 | 0 | 0 | 0.00013 |
| W2 | 0.8033 | 0 | 0 | 9.33 | 0 | 0 | 0 |
| W3 | 0 | 0 | 0 | | 0 | 0 | 0 |
| W4 | 0.37 | 0 | 0 | | 0 | 0 | 0 |
| W5 | 0 | 0 | 0 | 2.44 | 0 | 0 | 0 |
| W6 | 0.56 | 0 | 0 | 9.22 | 0 | 0 | 0 |
| W7 | 0.3933 | 0 | 0 | 0 | 0 | 0.008 | 0 |
| W8 | 0 | 0 | 0 | 0 | 0 | 0 | 0 |
| W9 | 0.3467 | 0 | 0 | 0 | 0 | 0 | 0 |
| W10 | 0 | 0 | 0 | 0 | 0 | 0 | 0 |
| W11 | 0 | 0 | 0 | 0 | 0 | 0 | 0 |
| W12 | 0.1867 | 0 | 0 | 0 | 0.0057 | 0 | 0 |
| W13 | 0.2333 | 0 | 0 | 0 | 0.0048 | 0 | 0 |
| W14 | 0.3267 | 0 | 0 | 0 | 0.0095 | 0.0032 | 0 |
| W15 | 0 | 0 | 0 | 0 | 0 | 0 | 0 |
| W16 | 0 | 0 | 0 | 0 | 0 | 0 | 0 |
| W17 | 1.3467 | 0 | 0 | 21.67 | 0 | 0.0019 | 0 |
| W18 | 0.110 | 0 | 0 | 2.0 | 0 | 0.0018 | 0 |
| W19 | 0.1230 | 0 | 0 | 2.0 | 0 | 0 | 0 |
| S1 | 0 | 0 | 0 | 0 | 0 | 0 | 0 |
| S2 | 0 | 0 | 0 | 0 | 0 | 0 | 0 |
| S3 | 0 | 0 | 0 | 0 | 0 | 0 | 0 |
| S4 | 0 | 0 | 0 | 0 | 0 | 0 | 0 |
| S5 | 0 | 0 | 0 | 0 | 0 | 0 | 0 |
| S6 | 0 | 0 | 0 | 0 | 0 | 0 | 0 |

**Table 1.** *Cont.*

| Station | Estrone | 17β-Estradiol | Estriol | Methyltestosterone | Stanozolol | Gestrinone | Cortisol |
|---------|---------|---------------|---------|--------------------|------------|------------|----------|
| R1 | 0.793 | 0 | 0 | 1.33 | 0 | 0 | 0 |
| R2 | 0.3 | 0 | 0 | 0 | 0 | 0 | 0 |
| R3 | 0.96 | 0 | 0.0257 | | 0.0021 | 0.0064 | |
| R4 | 0.853 | 0.56 | 0.029 | 2.56 | 0 | 0 | 0 |
| R5 | 0.603 | 0.08 | 0.0353 | 28.89 | 0 | 0.006 | 0 |
| A1 | 0.1267 | 0 | 0 | 0 | 0 | 0 | 0 |
| A2 | 0.1267 | 0 | 0 | 0 | 0.0154 | 0.0015 | 0 |
| A3 | 0.093 | 0 | 0 | 0 | 0 | 0 | 0 |
| A4 | 0.8267 | 0.087 | 0.0010 | 0 | 0 | 0.0019 | 0 |
| A5 | 0.6367 | 0 | 0 | 0 | 0.0045 | 0 | 0 |

Estrogen can cause feminization or monoecism phenomena in fish, resulting in reproductive problems. Simultaneously, the affected fish will be passed through the food chain and enriched in higher organisms, which could lead to an increase in the incidence of breast cancer and uterine cancer in women, and testicular cancer and prostate cancer in men, along with decreased sperm counts [43,44].

Estrone (E1) was the most frequently detected steroid in the water environment. The RQ value was 0 at the 13 stations where estrone was not detected, indicating no risk. However, the RQ value in the other detected stations ranged from 0.11 to 1.3467, indicating that E1 may cause low, moderate, or even high risk to aquatic organisms. In particular, the RQ value of station W17 in Tieshan Port Bay reached 1.3467, suggesting high risk. This station is close to municipal sewage outfall and the Baisha River estuary, with mariculture in the surrounding areas; seawater exchange capacity is weak in this area because of the location of the inner bay. Therefore, action should be taken to reduce the potential ecological risk to aquatic organisms, e.g., by improving the water quality in and around aquafarms, enhancing the governing of non-point source pollution around the Baisha River, and strengthening the management of municipal sewage treatment plants.

Methyltestosterone (MT) can be deleterious to reproductive development in aquatic organisms and mammals, as it interferes with normal endocrine function and affects the structure and function of the microbial community. It can inhibit gonadal development and reduce the reproductive ability of medaka at concentrations of more than 46.8 ng/L [45]. Many experiments have shown that the metabolism of MT is slow, with harmful characteristics that may be chronic, long term, and cumulative in humans. MT may interfere with the balance of natural hormones and disturb physiological functions in the human body. Its residue may cause acne, hirsutism, rough voice, amenorrhea, breast degeneration, changes in sexual desire, and other aspects of virilism in females. More seriously, it can affect the normal growth of children, even causing neonatal malformations [46]. MT is highly toxic: its *PNEC* is 0.09 ng/L, so even low concentrations indicate a high RQ value. The RQ values of all the stations with MT detected were greater than 1, indicating that they were all high-risk areas. The RQ value of the seawater sample of station W17 in Tieshan Port Bay reached 21.67, and the estuary station R5 reached a maximum of 28.89. MT was not detected in other studies, which means it is a steroid specific to this area.

17β-estradiol (E2β) was detected at three stations, with RQ values ranging from 0.08 to 0.56, considered as low or moderate risk. The RQ values for the other five steroids were less than 0.1, indicating no risk or low risk in these areas. Concerns have been raised that the risk assessment methods described above have some limitations. A key limitation is the difficulty to calculate the *PNEC* for each steroid because of the limited toxicity data, especially with the use of some synthetic growth promoters and progesterone. In this study, only free steroids were selected and used in the risk assessment; conjugate steroids were not selected, which may have led to an underestimation of steroid contamination in the environment.

3.3.2. Ecological Risk Assessment of Steroids in Marine and River Sediments

At present, there are little data regarding the toxicity of most hazardous organic pollutants in sediment. Moreover, the characteristics of sediment in different areas are very different (e.g., the content of organic carbon), which makes it difficult to accurately evaluate the risk. In addition, organic extraction and chemical analysis are often used to analyze the concentration of organic pollutants in environmental samples without considering bioavailability, as this may not be fully utilized by the organisms. However, there is a balanced action of organic pollutants that can convert the concentration of pollutants in the sediment into that of the pore water through the equilibrium distribution coefficient, so the ecological risk assessment of sediment can be carried out based on the *PNEC* value of the water. The concentration of pollutants in pore water can be deduced according to the following formula [40]:

$$C_{pore\ water} = \frac{C_{sediment}}{f_{OC} \times K_{OC}} \tag{3}$$

where $C_{pore\ water}$ is the concentration of pollutants in pore water (mg/L); $C_{sediment}$ is the concentration of pollutants in sediment (mg/kg); $K_{OC}$ is the normalized sediment/water distribution coefficient of the organic carbon in pollutants (L/Kg) (shown in Table 2); and $f_{OC}$ is the content of organic carbon of sediment (% (*m/m*)).

**Table 2.** *PNEC* values of target steroids (ng/L).

| Steroid | Estrone | 17β-Estradiol | Estriol | Methyltestosterone | Androsterone | Progestone | Cortisol | Prednisolone |
|---|---|---|---|---|---|---|---|---|
| *PNEC* | 3 | 1.5 | 60 | 0.09 | 100 | 545 | 2000 | 230 |
| $K_{OC}$(L/kg) [47] | 1047 | 794 | 372 | 372 | 562 | 2884 | 24 | 25 |

The results in Table 3 showed that the RQ values for 10 of the 17 stations detected with estrone were between 0.01 and 0.1, indicating that these stations were low-risk areas; only 1 station whose RQ value was greater than 0.1 but less than 1 had a medium risk. This is because estrone enters the ocean through rivers and sewage outlets, and its degradation rate is slow. After a long period of accumulation on the surface of sediments, the detected concentration of estrone is high. In addition, its *PNEC* is lower, thus resulting in a larger RQ value for estrone. Although estrone was at medium risk in this study, the highest concentration was only 270 ng/kg in W7, much lower than in previous studies of other regions [34,48,49]. The concentrations of the five other steroids, including 17β-estradiol, androsterone, progesterone, cortisol, and prednisolone, were all less than 0.1, indicating low risk.

**Table 3.** RQ values of target steroids in marine sediments.

| Station | Organic Carbon (%) | Estrone | | 17β-Estradiol | | Androsterone | | Progesterone | | Cortisol | | Prednisolone | |
|---|---|---|---|---|---|---|---|---|---|---|---|---|---|
| | | $C_{pw}$ | RQ | $C_{pw}$ | RQ | $C_{pw}$ | RQ | $C_{pw}$ | RQ | $C_{pw}$ | RQ | $C_{pw}$ | RQ |
| W3 | 0.86 | 0.033 | 0.011 | - | - | 0.0008 | 0.000008 | 0.00006 | 0 | - | 0 | - | - |
| W4 | 0.59 | 0 | 0 | - | - | 0.0024 | 0.00002 | 0.00005 | 0 | 0.006 | 0 | - | - |
| W5 | 0.33 | 0.55 | 0.18 | 0.0001 | 0.00008 | - | - | 0.00041 | 0 | - | 0 | - | - |
| W6 | 1.33 | 0.02 | 0.007 | - | - | - | - | 0.00008 | 0 | 0.0002 | 0 | - | - |
| W7 | 1.93 | 0.130 | 0.044 | - | - | - | - | 0.00009 | 0 | - | - | - | - |
| W8 | 1.08 | 0.034 | 0.011 | - | - | - | - | 0.00008 | 0 | - | 0 | - | - |
| W9 | 1.63 | 0.24 | 0.079 | 0.00004 | 0.00003 | - | - | 0.0002 | 0 | 0.0002 | 0 | - | - |
| W10 | 0.49 | 0.0001 | 0.00004 | - | - | - | - | 0 | 0 | 0 | 0 | 0.00001 | 0 |
| W11 | 1.79 | 0.046 | 0.015 | - | - | - | - | 0.00015 | 0 | - | - | - | - |
| W12 | 0.57 | 0.05 | 0.017 | - | - | - | - | 0.0001 | 0 | 0.003 | 0 | - | - |
| W13 | 0.26 | 0.125 | 0.042 | - | - | - | - | 0.0002 | 0 | - | - | - | - |
| W14 | 0.17 | 0.24 | 0.079 | - | - | - | - | 0.0002 | 0 | - | - | 0.01 | 0 |
| W15 | 1.21 | 0.022 | 0.007 | - | - | - | - | 0.00006 | 0 | 0.002 | 0 | - | - |
| W16 | 0.77 | 0.275 | 0.092 | 0.00004 | 0.00003 | - | - | 0.0002 | 0 | - | - | - | - |
| W17 | 0.77 | 0.045 | 0.015 | - | - | - | - | 0.0002 | 0 | - | - | - | - |
| W18 | 0.86 | 0.029 | 0.010 | - | - | 0.0008 | 0.000008 | 0.00006 | 0 | - | 0 | - | - |
| W19 | 0.59 | 0 | 0 | - | - | 0.002 | 0.00002 | 0.00005 | 0 | 0.006 | 0 | - | - |

Previous studies have shown that relatively high proportions and levels of steroid metabolites are omnipresent in the fishing port environment, finding that untreated municipal sewage and aquaculture waste could be the primary sources of steroid contami-

nation [50]. The entry of steroids into the aquatic environment poses a great ecological risk, and can derail endocrine hormones in wild animals [51]. Due to the influence of steroids, hermaphroditism and infertility have become common in fish. In this study, methyltestosterone and estrone were found to have a higher potential risk in Beihai Bay. Methyltestosterone is widely used in medicine, while estrone is used not only in medicine, but also in aquaculture and livestock farming. There have been many studies on the environmental effects of estrogens, including estrone. [3] Its characteristics of slow decomposition, continuous impact, and great harm should arouse our attention. Therefore, it is necessary to standardize its medical use, monitor estrone in sewage treatment plants, and strengthen environmental supervision.

## 4. Conclusions

(1) Overall, 9 out of 28 steroids were detected in the marine environment in the Beihai Bay area, and the main sources were seagoing rivers and the discharge of pollutants;

(2) Estrone and methyltestosterone were at high risk levels in the water environment of Beihai Bay. 17β estradiol (E2β) was at moderate risk, and other steroids were at low or no risk. Generally, ecological environmental risks in this region resulting in low or no detection are due to a low detection rate. However, attention should be given to these findings because the presence of various steroids can threaten aquatic organisms and human health;

(3) In conclusion, the pollution levels of steroids in water and the sediment environment of Beihai Bay were relatively light. Only one station had moderate risk due to estrone. However, with the rapid development of the marine economy and the growing population, and with the continuing expansion of aquaculture, steroids will be released at increasing levels into water environments through land surface runoff and mariculture, leading to uncertain potential risks in the future. Therefore, it is necessary to expand studies of the toxicological characteristics of steroids, especially conjugated steroids, so as to obtain more precise *PNEC* values for assessing the environmental risk of steroids more accurately.

**Supplementary Materials:** The following supporting information can be downloaded at: https://www.mdpi.com/article/10.3390/w14091399/s1, Table S1: Mobile phase conditions of liquid; Table S2: Basic information and UPLC MS /MS parameters for steroids and internal standards; Table S3: Recovery rate, detection limit and lower detection limit for steroids in ultrapure water and sediments by UPLC-MS /MS; Table S4: The content and distribution of steroids in water monitoring sections of seagoing rivers (ng/L); Table S5: The content and distribution of steroids in aquaculture wastewater; Table S6 The content, distribution, and removal efficiency of steroids at sewage treatment plants (ng/L); Table S7: Steroid content of seawater samples; Table S8: Steroid content of sediment samples (μg/kg).

**Author Contributions:** Conceptualization, H.Z. and C.R.; methodology, H.Z. and C.R.; software, C.R.; validation, H.Z. and C.R.; formal analysis, C.R. and X.T.; investigation, W.L. and C.H.; resources, H.Z. and C.R.; data curation, W.L. and C.H.; writing—original draft preparation, C.R. and X.T.; writing—review and editing, H.Z., C.R. and X.T.; visualization, H.Z.; supervision, H.Z.; project administration, H.Z.; funding acquisition, C.R. All authors have read and agreed to the published version of the manuscript.

**Funding:** This research was funded by National Natural Science Foundation of China, grant number 42076162, Guangdong Major Project of Basic and Applied Basic Research, grant number 2020A1515010496, and Guangxi Key Research and Development Program, grant number GuiKe AB20297018.

**Institutional Review Board Statement:** Not applicable.

**Informed Consent Statement:** Not applicable.

**Data Availability Statement:** Not applicable.

**Conflicts of Interest:** The authors declare no conflict of interest.

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
