# Peer review of "Sources, Pollution Characteristics, and Ecological Risk Assessment of Steroids in Beihai Bay, Guangxi"

_water, doi:10.3390/w14091399_

Round 1

Reviewer 1 Report

Journal: Water (ISSN 2073-4441) Manuscript ID: water-1661754 Type: Article Title: Sources, pollution characteristics, and ecological risk assessment of steroids in Beihai Bay, Guangxi Authors: Chao xing Ren, Xiao Tan, Wen lu Lan, Cui mei Huang, Hui Zhao
  Comments for authors:   The Abstract, Introduction, Materials and Methods, Results, and Conclusions sections were well explained with appropriate figures and tables in a way that readers can understand well.   However, I recommend the following modification for the paper before the publication.   Change the sub title of Results into Results and Discussion.  

Author Response

I'm sorry for my carelessness. We have changed the sub title of Results into Results and Discussion.  Thanks for your criticism. We have made other changes to the manuscript and look forward to your valuable suggestions.

Reviewer 2 Report

This study investigated the levels of steroids in various environmental matrices including seawater, sewage, and sediment in the Beihai Bay, Guangxi Province, and further assessed their ecological risks to the ecosystem.  One of my major concerns is that this work is relative descriptive and lacks of depth.  A new paragraph about environmental implication of this work may be added.  Besides, I would like to enrich the introduction section to clearly show the motivation of this work, i.e., why this study was necessary in the Beihai Bay?  These are already a lot of similar work relating to the detection and risk assessment of steroids.  However, no comparison was conducted between this work and previous studies.  There are some specific comments shown below that need to be addressed or clarified before the consideration for publication in the journal.

Specific comments

  1. Line 11. Six different types were detected for all monitored environmental matrices? Fig 3 shows 14 compounds detected in the sewage.
  2. Lines 78-93. How many replicates were set up for the sampling?
  3. Line 99. The internal standards were obviously lower than those used in Liu et al. (1 ppm, 100 µL).
  4. Lines 99 & 117. Which chemicals served as internal standards? How about the recovery and limit of quantification? No information is available in the manuscript.
  5. Line 100. Please check HLB is 600 mg or 500 mg?
  6. Lines 249-250. Change the sentence to “If sufficient toxicity data are available, the PNEC value can be derived using the method of species sensitivity distribution”.
  7. Line 270. From Table 2, E1 was the most frequent detected chemical?
  8. Table 3. Delete [Error!].
  9. Table 3. Did you derive these PNEC values?
  10. Table 4. I’m wondering if estrone has so high risk in sediment samples? Have you discussed and compared with those in previous studies?

Reviewer 3 Report

Please see in the attach.

Round 2

Reviewer 2 Report

This version is better than earlier one. However, there are still several concerns shown as below.

Line 70. Table S1 or Table S2?

Line 107. Change (500 mg, 100 mL) to (500 mg, 6 mL).

Line 303-304. The authors mentioned in line 303 that E1 was the most frequent detected steroid in the water environment. Why did line 304 say that estrone (E1) was not detected?

Line 386. I think at the moment there is no discharge guideline for estrone in sewage treatment plants. So it is not suitable to say to improve the discharge standard.

Table 4. If the authors have corrected Table 4, the whole body of the manuscript needs be carefully double checked. For example, in line 391 and elsewhere, E1 was at high risk level?

Table S3. Why were there three 10 ng/L?

Reviewer 3 Report

Accept in the present form.

Author Response

Thank you for your recognition and guidance.